# Prevalence and Correlates of Stunting among Children Aged 6–23 Months from Poor Households in Rwanda

**DOI:** 10.3390/ijerph20054068

**Published:** 2023-02-24

**Authors:** Jean de Dieu Habimana, Aline Uwase, Noel Korukire, Sara Jewett, Maryse Umugwaneza, Lawrence Rugema, Cyprien Munyanshongore

**Affiliations:** 1School of Public Health, College of Medicine and Health Sciences, University of Rwanda, Kigali P.O. Box 4286, Rwanda; 2School of Public Health, Faculty of Health Sciences, University of the Witwatersrand, Johannesburg 2193, South Africa

**Keywords:** Rwanda, stunting, children, poor, household, factors, intimate partner violence

## Abstract

Stunted linear growth continues to be a public health problem that overwhelms the entire world and, particularly, developing countries. Despite several interventions designed and implemented to reduce stunting, the rate of 33.1% is still high for the proposed target of 19% in 2024. This study investigated the prevalence and associated factors of stunting among children aged 6–23 months from poor households in Rwanda. A cross-sectional study was conducted among 817 mother–child dyads (two individuals from one home) living in low-income families in five districts with a high prevalence of stunting. Descriptive statistics were used to determine the prevalence of stunting. In addition, we used bivariate analysis and a multivariate logistic regression model to measure the strength of the association between childhood stunting and exposure variables. The prevalence of stunting was 34.1%. Children from households without a vegetable garden (AOR = 2.165, *p*-value < 0.01), children aged 19–23 months (AOR = 4.410, *p*-value = 0.01), and children aged 13–18 months (AOR = 2.788, *p*-value = 0.08) showed increased likelihood of stunting. On the other hand, children whose mothers were not exposed to physical violence (AOR = 0.145, *p*-value < 0.001), those whose fathers were working (AOR = 0.036, *p*-value = 0.001), those whose parents were both working (AOR = 0.208, *p*-value = 0.029), and children whose mothers demonstrated good hand washing practice (AOR = 0.181, *p*-value < 0.001) were less likely to be stunted. Our findings underscore the importance of integrating the promotion of handwashing practices, owning vegetable gardens, and intimate partner violence prevention in the interventions to fight child stunting.

## 1. Introduction

Stunted linear growth continues to be a public health problem that overwhelms the entire world and, particularly, developing countries. Stunting is a chronic undernutrition that affected 149.2 million children worldwide in 2020—22% of all children under five years old [1]. Countries of Oceania, excluding Australia and New Zealand, carry the most significant burden, with 41.4 % of children under five reportedly stunted, followed by Sub-Saharan Africa (SSA) with 36.8 % and South Asia with 31.8% [1]. Specifically, in Rwanda, stunting among children under five was 33.1% (8). Therefore, stunting is still among the main health problems that need more attention.

Stunting represents the cumulative effects of long-term undernutrition and repetitive infections during early childhood, mainly occurring in the first 1000 days of life [2]. It is defined as being too short compared to a child of the same age and sex or as children having a height-for-age Z-score less than minus two standard deviations based on the same population of reference [3,4].

The negative impact of stunting on child health is well documented. Early childhood stunting is associated with short stature [5], extending even into adolescence and adulthood [6]. It is also related to poor socio-psychological development [7], poor motor development, and reduced cognitive ability [8], as well as high risks of morbidity and mortality [9]. Stunting also contributes to low productivity later in life [4]. In addition, scientists have found that stunting in early childhood is associated with increased rates of obesity later in life [10]. Thus, adverse impacts of stunting attack the development of present generations, and harmful effects may manifest throughout their life journey.

The interrelation of sociodemographics and other factors contributes to the increased risks of stunting. It has been shown many times that poor socioeconomic status is a risk factor for stunting [11,12,13,14,15]. Additionally, stunting is associated directly with diseases and poor feeding practices. A systematic review showed that repetitive diarrheal episodes were associated with stunting [16]. Furthermore, poor feeding practice was positively associated with stunting [17]. Other factors that affect stunting are maternal nutrition status and education [18], as well as poor home environment [19]. As the factors associated with stunting are complex, there is a need to explore more factors.

Despite the political will and different programs and interventions that have contributed to the significant decrease in stunting in Rwandan children, the prevalence has not decreased as much as projected. However, there has been no study on the prevalence or risk factors of stunting in the poor households that are the most affected. Previous studies in Rwanda focused their interest on all socioeconomic categories using secondary data analysis from the Demographic and Health Survey (DHS) [15,20,21]. Others focused their attention on feeding practices only [17,22]. Thus, there is a shortage of studies that comprehensively explored these factors even though they could inform effective interventions. This study assessed the prevalence and factors associated with stunting among poor households of children aged 6–23 in Rwanda.

## 2. Materials and Methods

### 2.1. Study Setting and Population

A quantitative cross-sectional survey was conducted in the Rutsiro, Burera, Nyaruguru, Kayonza, and Gasabo districts from the Western, Northern, Southern, and Eastern Provinces and Kigali City, respectively. These districts were purposively chosen for their high stunting rates in their respective provinces and Kigali City based on the data from Rwanda Comprehensive Food Security & Vulnerability Analysis 2018 [23]. This study’s target population was children aged 6 to 23 months. In addition, we included mother–children dyads that belong to the family that has been identified as poor (Category 1 and 2 of Ubudehe) [24], children born full-term (between 38 weeks and 40 weeks), and singleton children (a child that is the only one born at one birth) in the study. Otherwise, eligible people who were very sick to the extent that there were not able to participate were excluded from the study.

### 2.2. Sample Size and Sampling Technique

A multi-stage cluster sampling approach was applied where the primary sampling unit was the administrative Village, followed by households. One mother–child dyad was selected from each household fulfilling the inclusion criteria of the survey. From each village, we selected five households systematically. A total sample of 877 mother–child dyads were recruited to take part in this study based on the formula for estimation of single population proportion $n=\frac{Z^{2}pq}{d^{2}}$ [25] where *n* is the desired sample size if the population is higher than 10,000, *z* is the x-coordinate of the standard curve that truncates a range at the ends if the confidence level is 95%, *z* = 1.96 *p* is the prevalence, and *q* = 1 − *p*. In this case, the prevalence was 33.1% at an accuracy of 5%. $n=\frac{1.96^{2}\left[0.331*\left(1-0.331\right)\right]}{0.05^{2}}=340.2$. We used a design effect of 2 and 20% to account for the non-response rate.

### 2.3. Study Variables

The primary outcome variable of this study was stunting, where children were categorized into stunted or not stunted. Explanatory variables included variables related to child characteristics such as age, sex, deworming status, Vitamin A supplementation, micronutrient powder supplementation, and minimum dietary diversity. Those variables were selected, referred to the existing literature, and considered for their socioeconomic and biological plausibility with stunting.

Household characteristics included the father’s employment status, household hunger status [26], household food insecurity access (HFIA) [27], household size, and owning a vegetable garden. Maternal characteristics included depressive syndrome [28], maternal employment status, maternal disability status, maternal literacy, maternal education, family planning type, breast discomfort during lactation, antenatal care visits, and mode of delivery. Other variables included intimate partner violence (IPV), which involves exposure to controlling behavior, emotional violence, physical violence, sexual violence, and any violence.

The survey also included questions related to Water, Sanitation, and Hygiene (WASH), including a source of drinking water, toilet facility, child stool disposal, handwashing facility, and observation of handwashing practice. In addition, we considered good handwashing practices, those who cut their nails and washed their hands with clean water and soap. WASH indicators were grouped and classified into improved and unimproved, following the WHO guidelines [29].

### 2.4. Data Analysis

To determine the prevalence of stunting among children aged 6–23 months from poor households, we calculated length-for-age z-scores using the WHO Anthro computer application. We then exported them in the Statistical Package for Social Science (SPSS) version 25.0 used for data analysis. Indices were categorized into stunted and not stunted based on the WHO 2010 Child Growth Standards, where stunting was defined as a Z-score less than −2SD and not stunted as a Z-score more than −2SD [30]. To calculate the prevalence of stunting, we divided the number of stunted children by the total number of children measured multiplied by a hundred. In descriptive statistics, we calculated frequencies and percentages for all variables. Additionally, we performed a bivariate analysis between stunting status and predicting variables. Due to the possible collinearity between independent variables, backward stepwise logistic regression was conducted to determine the final models. We included significant variables from bivariate analysis with *p* < 0.05 in the multiple logistic regression. In that process, variables with higher *p*-value (*p* > 0.05) and those which correlated with others were excluded automatically from the model, starting with the highest and stopping when all remaining variables were statistically significant (*p* ≤ 0.05). We reported the results as odds ratios (OR) with a 95% confidence interval (CI).

## 3. Results

### 3.1. Prevalence of Stunting

As shown in Figure 1, this study showed that the prevalence of stunting was 34.1%, which was a little bit higher than the national prevalence among those under five years old (33.1%).

### 3.2. Bivariate Analysis: Stunting and Background Characteristics

As shown in Table 1, 40.5% of children were between 6–12 months, 35% were between 13–18 months, and 24.4% were between 19–23 months. Additionally, 71.8% of children received the deworming tablet, 62.8% received micronutrient supplementation powder, and 92.3% received Vitamin A supplementation at six months old. Additionally, 82.2% of all children were breastfed within 1 h after birth. Results from this study show that 27.5% had received acceptable minimum dietary diversity.

Regarding maternal characteristics, 3.5% of mothers lived with a permanent disability, 21.2% were illiterate, and 11% had never gone to school. However, 86.2% of both parents have an income-generating activity, while 8.1% of fathers work alone in the households, and 5.7% of mothers are the only providers for the household. Further, 13.7% of mothers experienced breast discomfort during lactation. While 65.8% completed the standard antenatal care visits, nearly all (98.1%) received assistance from a health professional during delivery. Concerning IPV, 47.7% experienced controlling behavior from their husbands, 29.2% had experienced emotional violence, 27.3% reported physical violence, and 12.8% of all interviewed mothers reported sexual violence; overall, 57.3% experienced any type of violence. Over half of the mothers (46.6%) reported depressive syndrome.

In terms of household considerations, 8.1% of fathers were working alone. In addition, 71.5% of households experienced severe food insecurity, 17.8% demonstrated severe hunger, and 51.1% reported having a vegetable garden. About 29% of households fetched drinking water from an unimproved source, and 3.5% reported no toilet. Further, 4.4% had unimproved sanitation, and 5.5% reported inappropriate child stool disposal. In terms of WASH, 38.8% showed good handwashing practices.

In addition, from the same table, we measured the association between stunting and background characteristics. The following variables were significant (*p*-value < 0.05): child sex, child age, deworming, vitamin A supplementation, micronutrient supplementation, a mother living with disability, maternal ability to read and write, maternal education, breast discomfort during lactation, mode of delivery, controlling behavior from the husband, emotional violence from her husband, physical violence from her husband, sexual violence and any violence from her husband, father working alone, household hunger scale, child stool disposal, availability of the handwashing facility, handwashing practice, and owning a vegetable garden.

### 3.3. Multivariate Analysis: Factors Associated with Stunting

Table 2 presents logistic regression results to explore factors associated with stunting. Children aged 19–23 months were 4 times more likely to be stunted (AOR = 4.410, CI at 95% [1.911–10.173], *p*-value = 0.01) than those aged 6–12 months. Further, children aged 13–18 months were 3 times more prone to stunting (AOR = 2.788, CI at 95% [1.302–5.968], *p*-value = 0.08) compared to those aged 6–12 months. Additionally, mothers protected from physical violence were less likely to have stunted children (AOR = 0.145, CI at 95% [0.074–0.287], *p*-value < 0.001) than those exposed. Furthermore, households with fathers that have an income-generating activity and both parents have income-generating activity presented reduced odds of having stunted children, respectively, compared to those whose only mothers have an income-generating activity (AOR = 0.036, CI at 95% [0.005–0.242], *p*-value = 0.001; AOR = 0.208, CI at 95% [0.051–0.851], *p*-value = 0.029). Further, children whose mothers demonstrated good handwashing practices were less likely to be stunted (AOR = 0.181, CI at 95% [0.091–0.362], *p*-value < 0.001) compared to those who showed insufficient handwashing practice. Finally, children whose households did not have a vegetable garden were two times more likely to be stunted (AOR = 2.165, CI at 95% [1.201–3.905], *p*-value < 0.01) compared to those who have it.

## 4. Discussion

This study aimed to assess the prevalence of stunting and associated factors in Rwanda’s poor households with children aged 6 to 23. First, we assessed the prevalence of stunting based on the WHO indicators of Child Growth Standards. Despite efforts to reduce stunting since last decade, its prevalence was 34.1%, which was a little bit higher than the national prevalence [31] and far from achieving the target of 19% projected in 2024 [32]. Furthermore, children aged 6–23 are the most vulnerable to stunting [33]; this indicates the need for more efforts to address all risk factors, particularly among poor households [34].

The results from this study show that the odds of being stunted were higher among older children compared to the youngest. These findings are consistent with other studies conducted in Rwanda among children aged 0–59 months, where stunting increases as age increases [15,33]. For instance, a study conducted in Pakistan showed an increased risk of being stunted as age increased [35]. We think this could be attributed to the fact that some caregivers could give up and reduce efforts in caregiving as their children get older. It could also be explained by the weaning effect and progressive introduction to food that can interfere with some infections when there is poor hygiene [36].

It is also possible that children who received prolonged breastfeeding also tend to like it more and accept food with much more difficulty. This breastmilk addiction can lead children to poor nutrition and expose them to stunting [37]. Another explanation for stunting as age increases is that, as children grow, they start to explore their environment and eat many things on their way, exposing them to enteric infections, which cause environmental enteric dysfunction that can lead to stunting [38,39].

Children whose mothers were not exposed to physical violence were less likely to be stunted than those whose mothers were exposed. These results align with a study conducted in Bangladesh, where physical IPV was negatively associated with the linear growth of children [40]. In addition, another analysis of demographic and health surveys from 42 countries also demonstrated that maternal lifetime exposure to only physical violence was associated with childhood stunting, sexual violence, or both types combined [41]. Studies conducted in India and South Africa also found that physical violence was associated with increased risks of childhood stunting [42,43]. Scholars have explained various pathways by which physical violence can impair child growth. For example, poor psychosocial factors are believed to affect care practice, resulting in poor child growth [44]. In addition, long-lasting violence creates an environment of fear and poverty that leads to poor care practices and deprived nutrition statuses [45].

Apart from these factors that increase the risks of stunting among children, this study showed that when parents have income-generating activities, their children are mostly protected from stunting. These indicate both parents’ contributions to providing income that supports the household. This finding aligns with a study conducted in Rwanda that showed how paternal work contributes to another study in Ethiopia that showed that, as family incomes increase, there are reduced risks of stunted children [46]. Similarly, in Rwanda, a study identified a low wealth index as a risk factor for stunting [34].

This study showed that households whose father, alone, has income-generating activities tend to have fewer stunted children. These show that even though the mothers do not have a paying job, their contribution as housewives is still crucial for child growth. However, some studies have suggested that more involvement from men is not only in providing income but also in child caregiving, which could contribute to the better health of the children. For instance, in a study conducted in Kenya, maternal support from the husband and his participation in household decision-making positively affected the child’s linear growth [47]. Conversely, some studies conducted in Ethiopia and Indonesia showed that mothers working casually and in the agriculture sector tend to have more stunted children than homemakers and public sector workers [48,49]. In addition, mothers who invest more time in employment tend to offer limited follow-up time to their children. Consequently, insufficient time allocated to the child increases the risk of poor nutrition [50]. Thus, the mechanism of maternal empowerment should provide a window of interaction between mother and child.

Good handwashing practice was associated with decreased risks of having stunted children. This study explained good handwashing practice as combining washing hands with clean water and soap and cleaning nails. Like ours, many studies have shown the importance of WASH in the improvement of nutrition status. For example, a study in Guatemala clarified the connection between using soap, water, water availability, and diarrheal diseases [51]. Previous research demonstrated substantial shreds of evidence linking repetitive episodes of diarrheal diseases and stunting [52]. Therefore, stakeholders involved in improving hygiene practices should emphasize having primary materials, such as soap, and a better practice of hand washing.

The absence of a vegetable garden seemed to increase the risks of having stunted children. Similarly, a study in Lesotho found that owning a vegetable garden negatively affected stunting [53]. These risks could be due to a lack of food diversification at the household level and an inability to generate some additional income by selling surplus vegetables. However, another study in Rwanda found that owning a vegetable garden was positively associated with stunting, which was explained as a means of survival under hardship [54]. These risks might be explained, as well as the fact that having a vegetable garden does not guarantee good production or the better use of its products. More efforts should be made to ensure that vegetable garden is used to support household dietary diversity.

Firstly, our study was limited by the cross-sectional nature that did not allow us to make causal-effect inferences. Secondly, many factors assessed were self-reported without independent verification through records or observations, which could introduce biases, such as social desirability. Thirdly, there are limitations in the classification of wealth category. For example, the “ubudehe classification” does not scientifically determine the poverty level, which could bring bias since its classification can vary from one area to another.

## 5. Conclusions

The present study concluded that child age, physical violence experienced by mothers, absence of employment for both parents, absence of good hand washing practices, and absence of vegetable gardens were associated with higher levels of childhood stunting.

The nutrition intervention should focus on IPV mitigation, employment promotion for both parents, good handwashing practices, and vegetable garden ownership.

## Figures and Tables

**Figure 1 ijerph-20-04068-f001:**
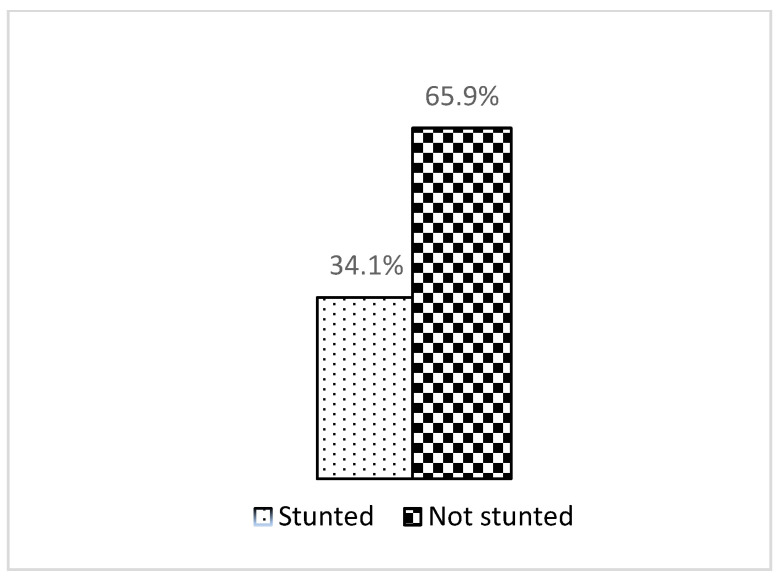
Stunting status.

**Table 1 ijerph-20-04068-t001:** Bivariate analysis: stunting and individual, family, and community-level factors.

Individual, Family, and Community-Level Characteristics	Stunting Status			*p*-Value
	Stunted % (N)	Not Stunted % (N)	Total % (N)	
Child sex				
Male	57.8 (159)	47.7 (254)	51.2 (413)	0.007
Female	42.2 (116)	52.3 (278)	48.8 (394)	
Child age in months				
6–12	27.6 (76)	47.2 (251)	40.5 (327)	<0.001
13–18	38.9 (107)	33.1 (176)	35.1 (283)	
19–23	33.5 (92)	19.7 (105)	24.4 (197)	
The child received deworming tablet in the last six months				
No	19.6 (54)	32.5 (173)	28.2 (227)	<0.001
Yes	80.4 (221)	67.5 (359)	71.8 (580)	
The child received Vitamin A supplementation at six months				
No	4.6 (12)	9.3 (45)	7.7 (57)	0.022
Yes	95.4 (247)	90.7 (438)	92.3 (685)	
The child received micronutrient supplementation regularly				
No	31.6 (87)	40.0 (213)	37.2 (300)	0.019
Yes	68.4 (188)	60.0 (319)	62.8 (507)	
Early breastfeeding after birth				
Within one hour	80.4 (217)	83.1 (433)	82.2 (650)	0.340
After one hour	19.6 (53)	16.9 (88)	17.8 (141)	
Child minimum dietary diversity				
No	71.6 (197)	72.9 (388)	72.5 (585)	0.696
Yes	29.4 (78)	27.1 (72)	27.5 (222)	
Mother lives with a disability				
No	94.9 (261)	97.6 (519)	96.6 (780)	0.047
Yes	5.1 (14)	2.4 (13)	3.5 (27)	
Mother can read and write				
No	25.5 (70)	18.0 (101)	21.2 (171)	0.033
Yes	74.5 (205)	81.0 (431)	78.8 (636)	
Maternal education				
No formal education	14.5 (40)	9.2 (49)	11.0 (89)	0.018
Primary	63.6 (175)	62.0 (330)	62.6 (505)	
Vocational, secondary, and high	21.8 (60)	28.8 (153)	26.4 (213)	
Family planning				
No	50.36 (138)	49.06 (260)	49.5 (398)	0.725
Yes	(49.64) 136	50.94 (270)	50.5 (406)	
Breast discomfort				
No	81.2 (225)	88.5 (471)	86.3 (696)	0.009
Yes	18.2 (50)	11.5 (61)	13.7 (111)	
Antenatal care visits				
less than 4	35.4 (97)	33.5 (177)	34.1 (274)	0.582
Four visits and higher	64.6 (177)	66.5 (352)	65.8 (529)	
Mode of delivery				
Not assisted by a health professional	3.6 (10)	0.9 (5)	1.9 (15)	0.007
Assisted by a health professional	96.4 (265)	99.1 (527)	98.1 (792)	
Family size				
Less than 5	33.1(91)	55.5 (295)	31.0 (250)	0.589
Over than 5	66.9 (184)	44.5 (237)	69.0 (57)	
Mother experienced controlling behavior from her husband				
No	37.1 (102)	60.2 (320)	52.3 (422)	<0.001
Yes	62.9 (173)	39.8 (212)	47.7 (385)	
Mother experienced emotional violence from her husband				
No	56.7 (156)	78.0 (415)	70.8 (571)	<0.001
Yes	43.3 (119)	22.0 (117)	29.2 (236)	
Mother experienced physical violence from her husband				
No	53.8 (148)	82.5 (439)	72.7 (587)	<0.001
Yes	46.2 (127)	17.5 (93)	27.3 (220)	
Mother experienced sexual violence				
No	77.8 (214)	92.1 (490)	87.2 (704)	<0.001
Yes	22.2 (61)	7.9 (42)	12.8 (103)	
Any type of violence from her husband				
No	26.9 (74)	50.9 (271)	42.7 (345)	<0.001
Yes	73.1 (201)	9.1 (261)	57.3 (462)	
Depressive syndrome (EPDS)				
No	51.6 (142)	54.3 (289)	53.4 (431)	0.468
Yes	48.4 (133)	45.7 (243)	46.6 (376)	
Parents working status				0.001
Mother only works	(8.4) 23	(4.3) 23	(5.7) 46	
Father only works	(4.0) 11	(10.2)54	8.1 (65)	
Both parents working	(87.6) 241	(85.5) 455	86.2 (696)	
Household food insecurity access (HFIA)				
Food secure and mild insecure access	4.7 (13)	7.5 (40)	6.6 (55)	0.158
Moderately Food Insecure Access	20.0 (55)	22.9 (122)	21.9 (177)	
Severely Food Insecure Access	75.3 (207)	69.5 (370)	71.5 (577)	
Household hunger scale (HHH)				
Little or no hunger	70.1 (108)	81.0 (239)	77.3 (347)	0.029
Moderate hunger	5.8 (9)	4.4 (13)	4.9 (22)	
Severe hunger	24.0 (37)	14.6 (43)	17.8 (80)	
Source of drinking water				
Unimproved	32.7 (90)	27.1 (144)	74.3 (234)	0.093
Improved	67.3 (185)	72.9 (388)	71.0 (573)	
Presence of latrine				
No	4.4 (12)	3.0 (16)	3.5 (28)	0.318
Yes	95.6 (263)	97.0 (516)	96.5 (779)	
Toilet facility				
Unimproved	4.6 (12)	4.2 (22)	4.4 (34)	0.847
Improved	95.4 (251)	95.7 (494)	95.6 (745)	
Child stool disposal				
Into latrine	90.2 (248)	96.8 (515)	94.5 (763)	<0.001
Elsewhere	9.8 (27)	3.2 (17)	5.5 (44)	
Availability of hand washing facility				
No	56.4 (155)	47.7 (254)	50.7 (409)	0.020
Yes	43.6 (120)	52.3 (278)	49.3 (398)	
Good handwashing practice				
No	43.7 (90)	16.9 (70)	25.9 (160)	<0.001
Yes	56.3 (116)	83.0 (342)	74.1 (458)	
Owning vegetable garden				
No	53.8 (148)	46.4 (247)	48.9 (395)	0.047
Yes	46.2 (127)	53.6 (285)	51.1 (412)	

**Table 2 ijerph-20-04068-t002:** Multivariate analysis: factors associated with childhood stunting.

Individual, Family, and Community-Level Characteristics	COR CI at 95%	*p*-Value	AOR CI at 95%	*p*-Value
Child’s Sex				
Female	1			
Male	1.500 [1.118–2.012]	0.007		
Child’s age				
6–12	1		1	
13–18	2.008 [1.413–2.854]	0.000	2.788 [1.302–5.968]	0.008
19–23	2.894 [1.979–4.230]	0.000	4.410 [1.911–10.173]	0.001
The child received deworming tablet in the last six months				
No	1			
Yes	1.972 [1.392–2.794]	0.000		
The child received Vitamin A supplementation at six months				
No	1		1	
Yes	2.115 [1.098–4.074]	0.025	3.638 [0.772–17.140]	0.102
The child received micronutrient supplementation regularly				
No	1		1	
Yes	1.443 [1.061–1.963]	0.020	0.597 [0.303–1.176]	0.136
Mother lives with a disability				
Yes	1			
No	0.467 [0.216–1.008]	0.052		
Mother can read and write				
No	1			
Yes	0.686 [0.485–0.971]	0.034		
Maternal education				
No formal education	1			
Primary	0.650 [0.412–1.025]	0.064		
Vocational, secondary, and high	0.480 [0.287–0.803]	0.005		
Breast discomfort				
Yes	1			
No	0.583 [0.388–0.875]	0.009		
Mode of delivery				
Not assisted by a health professional	1			
Assisted by a health professional	0.251 [0.085–0.743]	0.013		
Mother experienced controlling behavior				
No	1		1	
Yes	2.560 [1.897–3.456]	0.000	1.509 [0.809–2.817]	0.196
Mother experienced emotional violence				
Yes	1			
No	0.370 [0.270–0.506]	0.000		
Mother experienced physical violence				
Yes	1		1	
No	0.247 [0.178–0.342]	0.000	0.145 [0.074–0.287]	0.000
Mother experienced sexual violence				
No	0.301 [0.197–0.460]	0.000		
Yes	1			
Any type of violence				
No	1			
Yes	2.820 [2.056–3.869]	0.000		
Parents working status				
Only Mother works	1		1	
Only Father works	0.203 [0.085–0.485]	0.000	0.036 [0.005–0.242]	0.001
Both parents working	0.529 [0.291–0.963]	0.037	0.208 [0.051–0.851]	0.029
Household hunger scores				
Moderate hunger	1			
Severe hunger	1.243 [0.477–3.236]	0.656		
Little or no Hunger	0.653 [0.271–1.573]	0.342		
Child stool disposal				
Elsewhere	1		1	
Into latrine	0.303 [0.162–0.567]	0.000	0.303 [0.060–1.524]	0.148
Availability of hand washing facility				
Yes	1			
No	1.414 [1.055–1.895]	0.020		
Good handwashing practice				
No	1		1	
Yes	0.264 [0.181–0.384]	0.000	0.181 [0.091–0.362]	0.000
Owning vegetable garden				
Yes	1		1	
No	1.345 [1.004–1.801]	0.047	2.165 [1.201–3.905]	0.010

COR: Crude odd ratio, AOR, Adjusted odd ratio, CI: Confidence interval.

## Data Availability

The data will be accessible to anyone who desires to access them for scientific reasons. The request should be made through the corresponding author: jhabimana@cartafrica.org.

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
