# Peer review of "Prevalence and Correlates of Stunting among Children Aged 6–23 Months from Poor Households in Rwanda"

_ijerph, 2023, doi:10.3390/ijerph20054068_

Round 1

Reviewer 1 Report

Sentence 19 should read “associated factors of stunting, not with.

Sentence 21 and 89, define dyads

Rephrase sentences 24 to 32 on the results, there is too much information placed wrong that it makes following the thought harder. I suggest saying something like “children from households without gardens, and age (13-23 months) increased likelihood of stunting” then choose one measure to compare, maybe p value or CI, or all. Just make it easier to read and follow.

Sentence 41 should be rephrased for it is Asia that has the highest global rates of stunting followed by Africa. Make it clear your statement is referring to Africa alone not global.

Sentence 45, what agenda?

Sentences 47 to 51 are all definitions of stunting, reclassify one through definition and the z score one for diagnostic if you want to keep all of them.

You are overwhelming your sentences with too much information, break them down into a couple of stand-alone sentences so that they are easy to follow.

Sentences 55 to 59 have repeated impacts.

Sentence 64, grammar is past tense.

In sentences 124 to 135, you do not give the formula for prevalence calculation and the statistical software used.

Its often not recommended to repeat results, if it is reported in a table, do not report the exact thing in text, rather report something pertinent that is not in the table or give a summary. Table one results are repeated in sentences 138 to 160. Find a way to mitigate this repetition.

In your bivariate analysis table, I am not sure how to read the results as presented. For example on sex, you have reported that  p<0.007, but what is it comparing? Is it comparing stunted male vs stunted female i.e. 57.8% vs 42.2%, p<0.007 or non stunted male vs non stunted female? Or stunted male vs stunted female? Are males statistically more stunted than females? You may want to redo the table or clarify in text

It’s the same across the table. Make it clear what is being compared.

Sentences 181 to 185 and 187 to 192 need rephrasing, it’s not clear what you mean and poor grammar.

Table 3 is brilliant and clear. Make the other tables this clear and concise

Improve grammar in discussion section.

Sentences in 205 to 207 should state the ages reported in the studies so that it is not a repetition of the previous sentence and provide more substance to your findings.

Studies indicate that as children grow, they start to explore their environment and hence eat many things exposing them to infections which can lead to stunting. Research this idea and add it to your argument on sentences in lines 207 to 212.

Provide a bit more details on the 42 country study in line 216.

Rephrase sentence in lines 225 to 226, poor grammar.

Author Response

Dear Reviewer 1,

I have addressed your comments.

Thank you

Reviewer 2 Report

This good sample size (817) cross-sectional study has several strengths. The authors use the correct wording to refer to the “strength of the associations” and they include numerous important baseline predictor variables including: mothers exposure to physical violence parental employment, whether households had vegetable gardens, hunger and food insecurity, maternal depression, maternal disability status, maternal literacy, maternal education, breast discomfort during lactation, antenatal care visits, sources of drinking water, toilet facilities, child stool disposal and handwashing. They limited to full-term singleton births and they excluded those who are very sick, which is appropriate to avoid confounding by those variables. 

It is not clear how they used weight for height and weight for age to define stunting, as it is usually defined based on length for age, which they did include somehow in their definition.

A second weakness I can see in the study is that the authors did not define COR & AOR which are key in Table 3. It could be that they are referring to corrected and adjusted odds ratios but it is not at all clear. It could be that COR refers to the full model, and a OR refers to the reduced model to only variables that retains significance but it is not at all clear and need to be defined.

It would be helpful if table 1 was divided into two categories of stunted and not stunted to describe the sample well. Or, even better, if all the variables of table 1 are included in table too, table 1 should be simply omitted.

Author Response

Dear Reviewer 2,

I have addressed comment you provided

Thank you
